# Segment Anything Model for Instance Segmentation in Kidney Histopathology Images

**J.A.J. Weijer** [1]                                        J.A.J.WEIJER@UMCUTRECHT.NL

**S. Truijen** [2]                                            S.TRUIJEN-4@UMCUTRECHT.NL

**T.Q. Nguyen** [2]                                           T.Q.NGUYEN@UMCUTRECHT.NL

**M. Veta** [1]                                                      M.VETA@TUE.NL

**N. Stathonikos**[2]                             N.STATHONIKOS-2@UMCUTRECHT.NL

[1] *Technical University Eindhoven, The Netherlands*

[2] *University Medical Centre Utrecht, The Netherlands*

## Abstract

Kidney transplantation offers a significant improvement in the quality of life for individuals with irreversible kidney failure. Early detection of rejection through pathologists' assessment of kidney biopsies is critical to ensure long-term graft survival. Traditional assessment methods rely on semi-quantitative estimations from a pathologist while implementing deep learning models holds promise for providing more accurate measurements. Large annotated datasets, required to train such models, are challenging to obtain for kidney tissue. In this study, we fine-tune and modify the Segment Anything Model (SAM) to facilitate instance segmentation on whole-slide imaging (WSI) data. Leveraging SAM's zero-shot capability, we accelerate dataset creation by automatically obtaining annotations which we refine and label. We demonstrate promising results with limited annotated slides for training. Additionally, our approach allows for iterative dataset expansion to enhance model performance over time. Code is available at: https://github.com/JurreWeijer/SAM-Nephro.

**Keywords:** Multi-Class Segmentation, Segment Anything Model, Nephrology, Kidney Pathology, Kidney Transplantation

## 1. Introduction

Kidney transplantation significantly improves the quality of life for individuals with irreversible kidney failure (Garcia et al., 2012). The pathologists' assessment of protocol kidney biopsies after transplantation to detect early rejection is crucial to ensure long-term graft survival (Kers et al., 2022). Assessing the kidney tissue involves scoring using the Banff classification system by estimating factors like the degree of tubular atrophy/interstitial fibrosis (IFTA) and glomerulosclerosis (Roufosse et al., 2018) (Hermsen et al., 2019). Deploying deep learning models for autonomous tissue assessment can yield more accurate measurements compared to semi-quantitative estimations from a pathologist. Training such models requires large annotated datasets, which poses a challenge for kidney tissue because of the large number of small tissue features (Razzak et al., 2018). Foundation models offer a solution as fine-tuning for downstream tasks does not require an extensive dataset (Han et al., 2021). A notable foundation model for this problem is the Segment Anything Model (SAM), a zero-shot segmentation model that shows promise in accurately delineating image structures (Kirillov et al., 2023). However, SAM has not been trained on pathology images and does not provide labels for the masks.

This study aims to develop a deep learning model for automated renal allograft tissue assessment. We leverage SAM's zero-shot capability to speed up dataset creation by using its automatic mask generator to obtain unlabeled annotations that we refine and label. Using the dataset, we fine-tune a modified version of SAM to perform instance segmentation of the proximal tubules, distal tubules, atrophic tubules, glomeruli, sclerotic glomeruli, vessels, and background.

## 2. Methods

We gathered all slides stained with periodic acid-Schiff (PAS) from 72 patients containing tissue from protocol kidney biopsies taken three months after transplantation. The slides from 22 patients were used for training and validation, selecting one slide per patient, while we reserved the remaining 50 patients to evaluate histopathological features.

We developed a tool leveraging SAM and the automatic mask generator to speed up the annotation process. Slides were divided into patches for which masks were automatically generated, using overlapping patches to mitigate border artifacts. We employ a tissue mask to eliminate masks outside the tissue areas, while a size constraint ensures that masks do not exceed 4% of the slide patch area to exclude masks that are not of tissue features. When annotations in the patch overlap area did not agree, we kept the mask with the largest area when there was 90% overlap, otherwise, we kept the mask with the highest predicted IoU output from SAM. The segmentations are refined and labeled by a medical student, supervised by a pathologist, using QuPath (Bankhead et al., 2017). An "unspecified tubule" class was assigned when the class of tubular object could not be determined, which we omitted during the training process.

We maintained the majority of SAM's architecture but made some modifications to the mask decoder to obtain labels for the segmentations. We removed the option for multiple masks as output and introduced an additional class token. The class token is combined with the mask token, IoU token, and the prompt embeddings which, along with the image embedding, are the input for the two-way transformer. The output of the two-way transformer, resulting from class tokens, goes through an MLPs to obtain class probabilities as shown in Figure 1. We modified the annotation tool to accommodate labeled annotations for inference with the modified model. To obtain IFTA annotations, we accumulated clusters of adjacent atrophic tubules and created a hull around them.

## 3. Results and Discussion

The right side of Figure 1 displays the model's predictions compared to the ground truth. We highlight regions of glomerulosclerosis and atrophic tubules, as these are pivotal for clinical decision-making. We obtained the best performance for the glomerulus class with an mAP of 0.51, followed by proximal, distal, and atrophic tubules with mAP values of 0.43, 0.30, and 0.25, respectively. The sclerotic glomerulus and vessel classes exhibit the lowest performance, with mAPs of 0.08 and 0.13, respectively. The mAP is computed using a range of IoU values between 0.5 and 0.95 with an interval of 0.05 (Chen et al., 2023).

In Figure 2 on the left, we show the correlation between the degree of IFTA from the ground truth training data and the pathology report, which has a Pearson correlation coefficient of 0.43 (95% CI: 0.01-0.72), showing it is difficult to obtain a strong correlation. In Figure 2 on the right, we can see that the mean IFTA prediction of the model obtains a

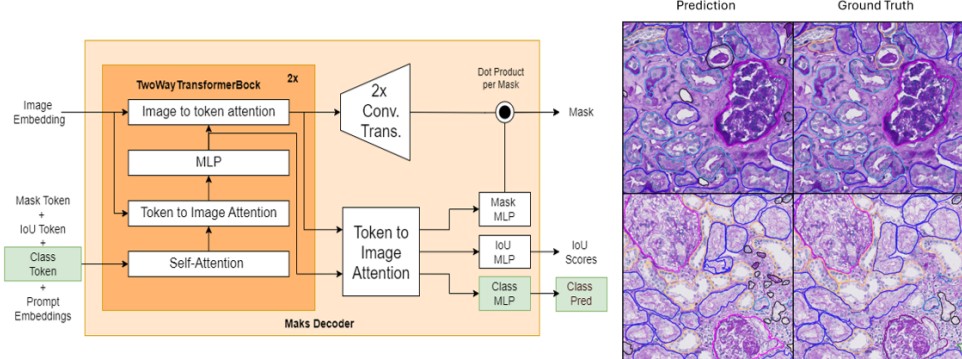

Figure 1: Left, the architecture of modified mask decoder, additions are displayed in green. Right, patches of predictions compared to the ground truth. The classes are: tubules proximal (blue), tubules distal (yellow), tubules atrophic (light blue), glomerulus (magenta), glomerulosclerosis (purple), vessel (red), and background (black)

similar correlation with a Pearson correlation coefficient of 0.42 (95% CI: 0.15-0.63). The predicted values are the mean of predictions made on three PAS-stained consecutive sections from the same biopsy of a patient.

For the training method, we do not require fully annotated slides which allows improvement of performance for certain classes by increasing the number of instances in the dataset. Similarly, we can expand the dataset with data from slides with more severe allograft pathology and differently stained slides to create a more robust model.

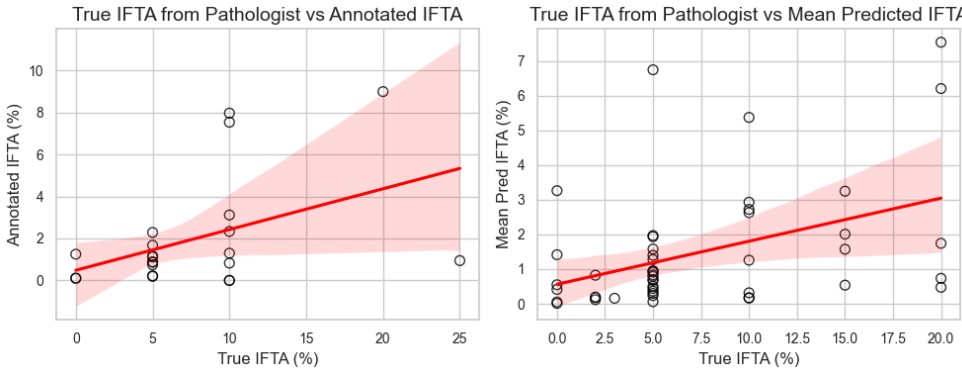

Figure 2: The correlation between the percentage of IFTA according to the pathology report and the percentage of IFTA from the ground truth (Left) or the mean prediction from the model (Right).

## 4. Conclusion

In this study, we generated a dataset using SAM, followed by modifying the model to facilitate instance segmentation of multiple tissue feature classes in kidney biopsies, aimed at assisting pathologists in their assessment of kidney biopsies. Our approach demonstrates promising results with a limited number of annotated slides for training, due to starting from a foundation model, while also having the flexibility to easily expand the dataset to enhance model performance in the future.

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
