# OpenReview forum: "Segment Anything Model for Instance Segmentation in Kidney Histopathology Images"
_MIDL.io/2024/Short_Papers — MIDL 2024 Short Papers_

### Official Review · Reviewer_Vqez · 2024-04-16

**Confidence:** 5
**Final Rating:** 4

**Review:**

The strength of this paper lies in its innovative approach to enhancing the analysis of kidney biopsies through the adaptation of the SAM for instance segmentation of multiple tissue features. The study leverages a foundational model to achieve promising results with a relatively small dataset. Additionally, the research design incorporates the flexibility to expand the dataset, which suggests potential for future improvements in model performance. These characteristics make the paper a valuable contribution to the field, showcasing a scalable and effective method for assisting pathologists in medical diagnostics.

---

### Decision · Program_Chairs · 2024-04-26

Accept